*Nat Genet.* Author manuscript; available in PMC 2022 June 18.

# Immune disease risk variants regulate gene expression dynamics during CD4+ T cell activation

**Blagoje Soskic**[#1,2],

**Eddie Cano-Gamez**[#1,2],

**Deborah J. Smyth**[1],

**Kirsty Ambridge**[1],

**Ziying Ke**[1],

**Julie C. Matte**[1],

**Lara Bossini-Castillo**[1],

**Joanna Kaplanis**[1,2],

**Lucia Ramirez-Navarro**[1],

**Anna Lorenc**[1],

**Nikolina Nakic**[3],

**Jorge Esparza-Gordillo**[3],

**Wendy Rowan**[3],

**David Wille**[3],

**David F. Tough**[3],

**Paola G. Bronson**[4],

**Gosia Trynka**[1,2]

[1]Wellcome Sanger Institute, Wellcome Genome Campus, Cambridge UK

[2]Open Targets, Wellcome Genome Campus, Hinxton UK

[3]GSK, R&D, Stevenage UK

[4]R&D Translational Biology, Biogen, Cambridge, MA, USA

[#] These authors contributed equally to this work.

## Abstract

During activation, T cells undergo extensive gene expression changes which shape the properties of cells to exert their effector function. Understanding the regulation of this process could help

Correspondence to: Gosia Trynka.

Correspondence to: Gosia Trynka (gosia@sanger.ac.uk).

**Author contributions**

GT, BS and ECG conceived and designed the project. BS, ECG, DJS and KA carried out the experimental work. BS, ECG, ZK, JCM and AL performed the data analysis. GT, BS, ECG, ZK, JCM, LBC, JK, LRM, NN, JEG, WR, DW, DFT and PGB interpreted the results. GT supervised the analysis. GT, BS, ECG, NN, JEG, WR, DW, DFT and PGB wrote the manuscript.

*Competing Interests:*All authors declare no competing interests.

explain how genetic variants predispose to immune diseases. Here we mapped genetic effects on gene expression (eQTLs) using single-cell transcriptomics. We profiled 655,349 CD4[+] T cells, capturing transcriptional states of unstimulated cells and three time points of cell activation in 119 healthy individuals. This identified 38 cell clusters, including transient clusters that were only present at individual time points of activation. We found 6,407 genes whose expression was correlated with genetic variation, of which 2,265 (35%) were dynamically regulated during activation. Furthermore, 127 genes were regulated by variants associated with immune-mediated diseases, with significant enrichment for dynamic effects. Our results emphasize the importance of studying context-specific gene expression regulation and provide insights into the mechanisms underlying genetic susceptibility to immune-mediated diseases.

## Introduction

Translating variants from genome-wide association studies (GWAS) to function provides insights into disease biology and improves treatment options[1]. Disease-associated variants from GWAS are enriched within active chromatin regions[2,3], implicating regulation of gene expression. These effects can be discovered using expression quantitative trait loci (eQTLs), which link variants to gene expression changes. However, most currently available eQTL maps use bulk tissues, and thus fail to capture gene expression dynamics, e.g. changes associated with a developmental stage[7,8] or external stimulus[9,10] in a given cell type[5] [6]. Mapping dynamic gene expression changes at single-cell level could overcome these limitations and provide insights into the molecular mechanisms underlying disease.

Variants associated with immune-mediated diseases are enriched in enhancers and promoters whose activity is upregulated upon CD4+ T cell activation[11,12]. However, CD4[+] T cells comprise naive cells, which have not yet encountered an antigen, and memory cells, which have previously undergone activation, both of which respond differently to activation[13-15]. Furthermore, memory cells consist of several subpopulations such as central memory ($T_{CM}$), effector memory ($T_{EM}$), and effector memory cells re-expressing CD45RA ($T_{EMRA}$), which differ in proliferative capacity and effector potential[16-18]. Additionally, regulatory T cells (Tregs), a subset of CD4+ T cells, control T cell activation and prevent excessive inflammation. Transcriptionally, these subpopulations form a continuum of phenotypes[19]. This cellular heterogeneity further complicates interpretation of immune disease-associated variants.

Given the dynamic nature of T cell activation and the heterogeneity of CD4[+] T cells, here we mapped gene expression regulation using single-cell transcriptomes spanning four timepoints of CD4+ T cell activation. We reconstructed activation trajectories for naive and memory CD4[+] T cells and identified eQTL effects manifesting at different timepoints and across different subpopulations of cells. We identified 127 genes with colocalizing eQTL and GWAS signals for immune-mediated diseases. Colocalizing genes were enriched in time-dependent eQTLs. Our data suggest that dysregulation of gene expression during T cell activation could underlie immune disease and emphasize the importance of context-specific gene expression regulation.

## Results

### Single-cell response of CD4+ T cells to activation

We isolated and stimulated naive and memory CD4+ T cells from 119 individuals and performed scRNA-seq[20] (Figure 1a, Supplementary Table 1 and 2, and Supplementary Figure 1a,b). We profiled cells in resting state, before dividing (16 h), after the first cell division (40 h), and after acquiring effector functions (5 d)[19]. This resulted in high quality data for 655,349 cells (Methods and Supplementary Figure 1c-g).

We performed dimensionality reduction and embedding using the uniform manifold approximation (UMAP)[21] (Methods) and observed that cells separated by timepoint of stimulation, forming a gradual progression from resting to the most activated cell state (cells collected at 5 d) (Figure 1b). This progression was accompanied by changes in activation markers. For example, an early activation marker, *CD69,* was upregulated at 16 h but downregulated at later timepoints, while expression of *IL2RA,* a marker of late activation, peaked at 40 h, remaining present at 5 d (Figure 1c). A population of cells localized at the intermediate point between resting and 16 h-stimulated cells (Figure 1b), and was composed of cells from the 16 h (74%) and 40 h (26%) timepoints. We hypothesized that this intermediate group represented an early activation state. By analyzing cells from these two timepoints independently, we observed that at each of these timepoints cells separated into two clear groups, one corresponding to the early activation state (Supplementary Figure 3). Cells in the early activation group expressed four-fold fewer genes compared to other cells at their respective activation timepoints and showed lower expression of T cell activation markers19 (Supplementary Figure 3). Furthermore, they showed a unique profile characterized by high expression of *STAT1, IFIT3,* and *GBP1* (Figure 1c). Therefore, these cells represent a distinct, early activation state, and we refer to them as lowly active.

Next, we performed unsupervised clustering of cells throughout the activation time course. This revealed a total of 51 cell clusters, which were merged into 38 cell populations based on their correlated patterns of gene expression (Supplementary Figure 4 and Methods). This included 25 stable subpopulations consistently detected at multiple timepoints, and 13 transient cell states only detected at specific timepoints (Figure 1d and Supplementary Table 3). Stable subpopulations belonged to one of five phenotypes: naive ($T_N$), central memory ($T_{CM}$), effector memory ($T_{EM}$), effector memory re-expressing CD45RA ($T_{EMRA}$), and regulatory ($nT_{reg}$) CD4+ T cells (Figure 1d,e). The memory pool consisted on average of 60% $T_{CM}$, 30% $T_{EM}$, 5% Tregs, and 5% $T_{EMRA}$ (Supplementary Figure 6a). The percentage of $T_{EM}$ cells decreased, while $T_{CM}$ and $T_{EMRA}$ increased with age (Supplementary Figure 6b). We observed no significant differences in subpopulations between sexes (Supplementary Figure 6c).

Additionally, we observed transient cell states that were only detected at specific activation timepoints (Figure 1d,e), for example, a population of cells expressing high levels of interferon (IFN)-induced genes (eg. *IFI6, IFIT3, ISG15, MX1)* during early activation (Supplementary Figure 5). Another subpopulation expressed high levels of NF-κB response genes (eg. *NFKBID, REL, BCL2A1*) (Supplementary Figure 5) and was dominant at mid-stages of activation. Additionally, during late activation we observed a population of

mitotic cells and a group of cells expressing high levels of heat shock protein (HSP) family members (e.g. *HSPA1A, HSPA1B, DNAJB1;* Supplementary Figure 5). Notably, HSPs have been implicated in controlling T cell responses to fever[22]. We also observed a subset of $T_{EM}$ cells which upregulated HLA molecules (eg. *HLA-DRA, HLA-DPA1, HLA-DRB1*) during late activation (Supplementary Figure 5). Importantly, all individuals contributed uniformly to each cluster, with more variability observed in $T_{EMRA}$, as previously described[17,24] (Supplementary Figure 5f).

## A temporal eQTL map of CD4[+] T cell activation

To study the genetic regulation of gene expression throughout T cell activation, we performed cis-eQTL mapping. For each timepoint, we reconstructed average transcriptional profiles per cell type and individual (i.e. pseudobulk transcriptomes) corresponding to $T_N$ and $T_M$ CD4+ T cells (Methods). We detected 1,545-3,006 genes with significant *cis*-eQTL effects (eGenes) at different activation timepoints (Figure 2a), of which 210-640 eGenes were only detected in individual cell states (Figure 2b). For example, the kinase gene *NME4* and purinoceptor gene *P2RX4* only showed effects in $T_M$ at 16 h and 40 h of activation, respectively (Figure 2c). The multivariate adaptive shrinkage (mashR) method[25] revealed a higher level of eQTL sharing across cell types within the same timepoint (Supplementary Figure 7a) than across different timepoints, suggesting that eQTL effect sizes change throughout activation. We also observed a high replicability (0.67-075) of our results with publicly available CD4+ T cell eQTLs from bulk RNA-seq[26,27] (Supplementary Figure 7c). However, eQTL sharing was reduced when taking into account both the direction and the magnitude of eQTL effects (0.28-0.34); Supplementary Figure 7c), suggesting that effect sizes might differ between different transcriptomic profiling strategies, naive and memory cells, and across T cell activation timepoints.

To gain a more granular view of gene expression regulation throughout T cell activation we mapped eQTLs in the 38 cell populations (Figure 1). As expected, we observed a high overlap between eGenes detected in different subpopulations (Figure 2d). Nevertheless, $T_{EM}$ and cells expressing HLA genes ($T_{EM}$ HLA[+]) had a higher number of specific eGenes (62-97%) compared to other populations, suggesting that they are more transcriptionally different than other subsets. Small subpopulations, such as $T_{EMRA}$, yielded a low number of eGenes (3-23) which suggested that the statistical power to detect eGenes correlates with the number of cells profiled ($R^2 = 0.82$, $p = 4.8 \times 10^{-10}$) (Figure 2e). Indeed, when we subsampled different numbers of $T_{CM}$ cells and repeatedly performed eQTL mapping we observed that eGene discovery increased proportionally to the number of cells analyzed (Supplementary Figure 7b). Despite this, we identified subpopulation-specific eGenes absent from the whole $T_N$ or $T_M$ populations. For example, 56-153 eGenes (10%-16%) were found in the subpopulation of naive cells characterized by expression of high levels of IFN-induced genes, but these effects were absent from the whole activated $T_N$ cells (Figure 2f). Similarly, 47-528 (13-31%) eGenes were detected in either of the two largest $T_M$ subpopulations ($T_{CM}$ and $T_{EM}$) but not in the whole $T_M$ population (Figure 2g). For example, *GNPDA1* was an eGene in $T_{CM}$ but not in $T_{EM}$ or in whole memory cells (Figure 2g). These genes were only detected as eQTLs in specific cell clusters and we observed that many were not detected in the Database of Immune Cell eQTLs (DICE) dataset which includes different subsets of

CD4+ T cells in resting and stimulated states. For example, *VAMP8 and AIMP1* eQTLs ($T_{CM}$ specific $p_{adj}$ = 5 x 10$^{-4}$ and $p_{adj}$ = 1.04 x 10$^{-4}$, respectively) and *RNF168* (specific to IFN-expressing cell cluster $p_{adj}$ = 6.2 x 10$^{-3}$), were not detected in any of the T cell populations in DICE. Therefore, as more studies emerge the power to detect cluster-specific eQTLs will increase, uncovering eQTLs that were previously undetected in bulk tissues.

## Cell-type specific co-expression gene modules

We next sought to understand which transcriptional programs shape the T cell response to activation, and whether eGenes regulate T cell functions. We computed pairwise gene expression correlations[28] of 11,130 highly expressed and variable genes, across 106 individuals and the 38 identified cell populations (Figure 3a, Supplementary Figure 8 and Methods). We identified 12 gene modules which represent key cellular functions involved in T cell activation (Figure 3b and Supplementary Table 4 and 5). For example, module 4 contained genes involved in the regulation of cell cycle checkpoints and DNA repair, and was highly expressed at 40 h and 5 d after activation, consistent with the timing of the first cell division[29]. Furthermore, module 11 included genes whose expression peaked in lowly active and 16 h-stimulated cells and remained high at later timepoints. These genes were involved in IFN-induced antiviral mechanisms such as OAS and ISG15-signalling, which are induced rapidly upon viral infection.

In addition to separating genes by temporal dynamics, the co-expression networks also highlighted subpopulation-specific gene expression modules, corresponding to effector T cell functions. For example, genes involved in cytokine secretion and interleukin signaling were highly expressed in $T_{EM}$ and $T_{EMRA}$, but not $T_{CM}$ or $T_N$ cells (Figure 3a,b), reflecting the potential of $T_{EM}$ and $T_{EMRA}$ cells to respond quickly[18,19]. Consistent with this observation, $T_{EM}$ and $T_{EMRA}$ showed high expression of TCR-induced genes (i.e. targets of ZAP-70 and downstream of CD3 zeta chain phosphorylation) at an earlier stage of activation, while other subpopulations did not express these genes until 40 h after stimulation (Figure 3a,b). Furthermore, we observed that module 12, which included genes important for cytotoxic function and chemokine signalling, was most highly expressed in $T_{EMRA}$ (Figure 3a). This cytotoxic capacity distinguishes $T_{EMRA}$ from other T cell subpopulations.

Next, using a permutation strategy (Methods), we showed that eGenes detected in activated T cells were particularly enriched in modules 2 (metabolism), 3 (cell division) and 9 (immune processes) (Figure 3c). In contrast, eGenes detected in resting cells showed strongest enrichment in module 6 (RNA metabolism, Herpes infection) (Figure 3c). Finally, we observed that eQTL effect sizes, as well as log-transformed allelic fold-changes [30], negatively correlated with the centrality values of their corresponding eGenes in the coexpression network, i.e. eGenes with larger eQTL effects were less connected in the network (Figure 3d and Supplementary Figure 8b,c). This suggests that genes at the edges of the co-expression network are more tolerant to variation in gene expression.

## Modelling of time-dependent eQTL effects

Previous studies showed that eQTLs can be context-specific[9,31]. Therefore, we assessed the role of genetic variation on the regulation of gene expression dynamics throughout T cell activation (dynamic eQTLs). We used trajectory inference[32] (Methods) to model activation time as a continuous variable (Figure 4a). The inferred trajectory agreed with the timepoints profiled experimentally and T cell activation markers such as *IL7R* (reduced expression upon activation), *CD69* (early activation) and *IL2RA* (early and late activation) (Figure 4b) also followed their expected expression patterns. In total, we identified 5,090 genes for which expression changed as a function of pseudotime (Supplementary Table 6). For example, *IRF1* and *TOP2A* were respectively downregulated and upregulated at late stages of activation (Figure 4b). Dynamically regulated genes were enriched in pathways related to T cell activation, such as DNA replication and regulation of cell cycle, mRNA transcription and processing, protein translation, signalling downstream of the TCR, and signaling by interleukins (Supplementary Table 7). Finally, we observed that $T_M$ were characterized by lower pseudotime values than $T_N$ sampled at the same timepoints. This is a consequence of $T_M$ showing a shorter activation path, likely reflecting a faster response.

To model dynamic eQTLs, we divided the pseudotime trajectory into ten bins and averaged the expression of genes per individual in each bin (Methods). Splitting the trajectory enabled us to control for the numbers of cells and therefore to reliably estimate mean gene expression values. We then used mixed models to identify eQTLs for which the effect size changed as a function of activation time (Figure 4c and Methods). We identified 2,265 genes with dynamic eQTL effects, which comprised 34% of eGenes in our dataset (Supplementary Table 8 and Supplementary Figure 9a). We used a permutation-based strategy to validate that this method was well calibrated (Methods and Supplementary Figure 9b). We applied both linear and quadratic models and observed that most eQTLs followed linear dynamics (74% and 76% in $T_N$ and $T_M$ cells, respectively; Figure 4e). However, for 502 and 495 genes in naive and memory cells respectively, we detected a non-linear interaction with activation pseudotime. For example, *GBP7* and *CFLAR* demonstrated eQTL effects only upon activation and their magnitude peaked at mid stages of the pseudotime trajectory (Figure 4d). In contrast, the magnitude of an eQTL for *SERINC5* peaked at early stages of the trajectory and diminished throughout activation (Figure 4d), while an eQTL for the interferon alpha inducible gene *IFI27L1* showed an effect size which linearly increased along the activation trajectory.

Finally, linear eQTLs were enriched in metabolic pathways, while non-linear eQTLs were enriched in both metabolic and immune processes (e.g. T cell proliferation and leukocyte degranulation) (Figure 4f). This suggests that for many immune genes, genetic regulation is only evident during specific stages of T cell activation.

## Colocalization at GWAS loci identifies immune disease genes

We obtained summary statistics for 13 immune-mediated diseases available in the GWAS catalog[33] (Methods) and tested for colocalization[34 35] (Methods) with the eQTLs mapped to $T_N$, $T_M$, and the subpopulations. We identified 471 unique colocalizations (PP4 > 0.8), corresponding to 247 GWAS loci for 11 diseases and 314 SNP-gene pairs (Supplementary

Table 9 and 10). This enabled us to prioritize 127 candidate disease-causal genes (Figure 5a). Importantly, 77 (60%) colocalizing genes were detected upon activation, and would have been missed by profiling only steady state *ex vivo* cells. Out of those, 47 (37%) were captured specifically in later timepoints of activation (40 h + 5 d) (Figure 5b). This is important, as previous eQTL studies have relied on either resting cells or a single, usually early activation timepoint[2,27].

Generally, we observed more colocalizations in larger cell populations (for which we were more powered to detect eQTLs) and in traits with larger numbers of reported GWAS signals (Figure 5a). The traits with the highest number of colocalizations were Crohn's disease and ulcerative colitis, followed by allergic diseases (AllD), in agreement with their proposed T cell-driven biology[11,12,36]. Nevertheless, higher number of colocalizations was not only a consequence of more powered GWAS. Systemic lupus erythematosus (SLE), although characterized by a higher number of loci compared to type-1 diabetes (T1D), had a smaller proportion of colocalizing variants, in line with studies pointing towards B cells as drivers of SLE[11,37]. We found that 72% of genes colocalized only with one trait, 14% with two traits, and 14% with 3 or more diseases (Supplementary Figure 10a). Overall, 220 disease loci (89%) regulated a single gene, while 22 (9%) and 5 (2%) loci regulated two and three genes in the associated regions, respectively.

While most colocalizing genes were detectable in broad cell types (i.e. total $T_N$ or $T_M$ per timepoint; median per trait = 66%), we observed between 2 and 15 genes per disease (median per trait = 25%) which were only detected in individual subpopulations (Figure 5c). For example, an eQTL for *TYK2* specifically detected in 16 h-stimulated $T_{EM}$ cells colocalized with a Crohn's disease association (Supplementary Figure 10b). Similarly, we identified a colocalization between a Crohn's disease locus and a *ZMIZ1* eQTL specific to 16 h-stimulated $T_{CM}$ cells (Supplementary Figure 10c). This eQTL is absent in other memory T cell populations such as $T_{EM}$. This leads to the eQTL being masked in bulk memory cells, where it is no longer detectable (Supplementary Figure 10c). Both of these colocalizations are subpopulation and timepoint-specific, which highlights the importance of measuring gene expression regulation with cell type and state resolution. We observed no differences in the network connectivity of colocalizing genes compared to other eGenes (Supplementary Figure 10d).

Given that the majority of colocalizations were detected in activated T cells we asked if these genes showed dynamic genetic regulation. Dynamic eQTLs were enriched in colocalizing eGenes in both naive and memory T cells (36/73 and 44/72 colocalizing genes in naive and memory cells, Fisher's test p-values 7.9 x 10$^{-5}$ and 2.6 x 10$^{-7}$, respectively). The expression patterns of most colocalizing eGenes were similar between naive and memory cells (Figure 5d). An example of a gene whose genetic regulation differs between naive and memory cells is the gene encoding the IL-18 receptor (IL18R1), a dynamic eQTL in memory T-cells. IL18R1 is highly expressed during early activation of memory cells and, conversely, during late activation of naive cells (Figure 5e). Another example is *CTLA4,* a dynamic eQTL in both memory and naive T cells, but with different regulation in the two cell types (Figure 5f); naive cells upregulated and maintained high expression of *CTLA4* upon activation, while memory cells highly expressed *CTLA4* only during early activation.

This eQTL colocalized with a T1D-associated locus and individuals carrying the disease risk allele showed lower expression of *CTLA4*. Reduced expression of *CTLA4* at early stages of activation could result in impaired ability to suppress T cells, thus contributing towards excessive activation in disease. Additionally, the same eQTL variant colocalized with association signals for rheumatoid arthritis and celiac disease, in agreement with the CTLA4-based therapies used in rheumatoid arthritis[38] (Supplementary Table 9).

Finally, we asked if immune disease loci affected specific cellular functions. Colocalizing genes were enriched in pathways involved in the regulation of T cell activation and proliferation (Figure 5g). There were 26 genes driving this enrichment, including genes with association signals shared across two or more diseases. For 24 out of 26 genes the direction of effect of the risk allele on gene expression was consistent between traits. Colocalizing genes also clustered into connected modules based on the information in STRING[39], i.e. the genes were co-expressed across tissues or proteins they coded for were physically interacting (Figure 5h). Furthermore, neighboring genes within these modules tended to be perturbed in the same direction by immune disease variants. For example, we observed a module of interconnected genes, 12 of which were involved in the regulation of T cell activation and proliferation. Among these, *PTPRC* was directly connected to *CD6, CD5, CTLA4,* and *TNFRSF14.* Notably, all of these genes were downregulated by risk alleles, suggesting that their reduced expression may increase disease risk. Our results demonstrate that immune disease loci colocalize with genes involved in the regulation of T cell activation, and that genes with similar functions tend to be perturbed in the same direction by disease risk alleles.

## Discussion

Dysregulation of T cell activation can result in poor response to infections, development of inflammatory diseases, or primary immunodeficiencies. By using single-cell profiling across 655,349 CD4[+] T cells, our study provides an unbiased view of the T cell response to activation, revealing 38 distinct subpopulations. This single-cell resolution provides new explanations to previous results from bulk gene expression. For example, we recapitulated the up-regulation of IFN-related genes early upon CD4+ TCR engagement[40] and further resolved it to a specific subpopulation of naive cells. We also demonstrated that the previously described modulation of HLA molecules upon T cell activation[23] is driven by $T_{EM}$ cells. Therefore, our data provide a resource for the interpretation of studies of T cell function.

Often eQTLs obtained from bulk RNA-seq mask cell-type specific effects[41], which can be mapped with single-cell transcriptomics[42]. Many immune cell eQTL resources[26,27], including those capturing T cell activation[40], rely on sorting cells based on surface markers. However, these approaches cannot capture the full cellular heterogeneity. Here, scRNA-seq allowed us to map eQTLs within clusters unbiasedly, providing insights into genetic regulation in different cell subsets. Our study will help infer the effects of genetic regulation on the development of effector T cell functions, and could inform cell engineering approaches.

eQTLs can be context-specific, including those resulting from responses to stimuli[9,31]. However, current eQTL resources mostly include cells in steady state. While these resources are instrumental in interpreting GWAS signals, the proportion of GWAS-eQTL colocalizations remains low[43]. In contrast, our study captured context-specific gene expression regulation. In particular, had we only focused on the resting state, we would have missed most disease-relevant eQTLs, as only 40% of colocalizations are detectable in resting cells. Furthermore, colocalizing eQTLs were enriched for eGenes with dynamic regulation, which could explain why at present eQTLs have only explained a small proportion of GWAS associations.

Finally, our results could inform drug target discovery. For example, a *CTLA4* eQTL colocalizes with GWAS associations for three immune diseases, where the disease risk alleles decrease gene expression. CTLA4 removes costimulatory molecules from the surface of antigen presenting cells, downregulating T cell activation[44]. Thus, a partial reduction in CTLA4 function could impair immune regulation and increase the risk of autoimmunity[45]. This is supported by existing therapies where a CTLA4 fusion protein is administered to patients with rheumatoid arthritis to help reduce inflammation[46]. Importantly, we show that the expression of *CTLA4* is dynamically regulated, peaking during early activation. Similarly, a *TYK2* eQTL detected in $T_{EM}$ cells colocalizes with a Crohn's disease GWAS association. The *TYK2* locus is associated with 10 different immune disorders, with three independent signals reported[1,47,48]. One of these signals is explained by a missense variant, which reduces signalling downstream of several cytokine receptors, resulting in protection from disease[1].

Here, we show a similar effect, where individuals carrying a protective allele for Crohn's disease have lower expression of *TYK2* in $T_{EM}$ cells at 16 h of activation. Inhibition of TYK2 as a treatment for inflammatory diseases is in clinical trials[49]. These examples illustrate how colocalizing genes could have therapeutic value.

We note that a limitation of our study is that we profiled healthy individuals. While this enabled us to identify eQTLs involved in disease susceptibility, we are likely missing eQTL colocalizations relevant for disease progression. Future studies in disease cohorts will be required to understand genetic regulation after disease onset.

## Methods

### Cell isolation and stimulation

Blood samples were obtained from 119 healthy individuals of British ancestry. Of these, 67 were male (53.7%) and 52 female (56.3%), and the mean age of the cohort was 47 years (sd = 15.61 years) (Supplementary Figure 1a). Human biological samples were sourced ethically and their research use was in accord with the terms of informed consent under an IRB/EC approved protocol (15/NW/0282). Peripheral blood mononuclear cells (PBMCs) were isolated using Ficoll-Paque PLUS (GE healthcare, Buckingham, UK) density gradient centrifugation. Naïve (CD25- CD45RA+ CD45RO-) and memory (CD25- CD45RA- CD45RO+) CD4+ T cells were isolated from the PBMC fraction using EasySep® naïve CD4+ T cell isolation kits and memory CD4+ T cell enrichment kits

(StemCell Technologies, Meylan, France) according to the manufacturer's instructions. Naive and memory T cells were then stimulated with anti-CD3/anti-CD28 human T-Activator Dynabeads® (Invitrogen) at a 1:2 beads-to-cells ratio. Cells were harvested after 16 hours, 40 hours, and 5 days of stimulation. In addition, unstimulated cells kept in culture without any beads for 16 hours were used as a negative control (i.e. zero hours of activation).

### Single-cell RNA-sequencing

Upon harvesting, cells were resuspended in RPMI media to obtain a single-cell suspension. Next, cells were stained with the live/dead dye 4',6-diamidino-2-phenylindole (DAPI) and dead cells were removed from the suspension using fluorescence-activated cell sorting (FACS). Live cells were resuspended in phosphate buffer saline (PBS), at which point cells obtained from different individuals but belonging to the same experimental condition were mixed together at equal ratios to form a single cell suspension (i.e. pool). Each pool corresponded to a mix of cells from four to six different individuals (median = 6), and we processed a total of 172 pools.

Cells were next processed for single-cell RNA-sequencing using the 10X-Genomics 3' v2 kit[20], as specified by the manufacturer's instructions. Namely, 1 x 10$^4$ cells were loaded into each inlet of a 10X-Genomics Chromium controller in order to create GEM emulsions. Each experimental condition was loaded in a separate inlet. The targeted recovery was 6,000 cells per pool. Reverse transcription was performed on the emulsion, after which cDNA was purified, amplified, and used to construct RNA-sequencing libraries. These libraries were sequenced using the Illumina HiSeq 4000 platform, with 75 bp paired-end reads and one cell pool per sequencing lane.

### Genotyping

Genomic DNA was isolated from a suspension of 1 x 10$^6$ PBMCs from each individual in the study using a DNA isolation kit (Qiagen). Genotyping was then performed using the Infinium CoreExome-24 (v1.3) chip (Illumina). Genotype data were analyzed as detailed in Supp. Note.

### Single-cell RNA-sequencing data analysis

**Data processing and quality controls**—Raw scRNA-seq data were processed using the Cell Ranger Single-Cell Software Suite[20] (v3.0.0, 10X-Genomics). In brief, reads were first assigned to cells and then aligned to the human genome using STAR[55], with the hg38 build of the human genome (GRCh38) as a reference for alignment. Ensembl (v93) was used as a reference for gene annotation, and gene expression was quantified using reads assigned to cells and confidently mapped to the genome.

Results from RNA quantification in Cell Ranger were imported into Python (v3.8.1) and analyzed using scanpy (v1.4.4)[56]. Samples with less than 70% of reads mapping to cells were discarded. This resulted in 142 (82%) cell pools and 106 (89%) individuals being kept after quality filters. In addition, any cells with fewer than 200 detected genes, an unusually high number of genes (defined as over four standard deviations above the mean number of detected genes), or more than 10% of reads mapping to mitochondrial genes were removed

from the data set. Finally, any genes detected in fewer than 10 cells were discarded. This resulted in 713,403 cells (96.77% of total) and 23,360 genes passing quality filters.

**Deconvolution of single cells by genotype—**Each scRNA-seq sample comprised a mix of cells from unrelated individuals. Thus, natural genetic variation was used to assign cells to their respective individuals. First, a list of common exonic variants was compiled from the 1000 Genomes Project phase 3 exome-sequencing data[53]. This list included any variants with a minor allele frequency of at least 5% in the European population. Next, cellSNP (v0.99)[57] was used to generate pileups at the genomic location of these variants. These pileups, in combination with the variants called from genotyping in each individual, was used as an input for Vireo (v1)[57]. Vireo uses a Bayesian approach to infer which cells belong to the same individual based on the genetic variants detected within scRNA-seq reads. Any cells labelled as "unassigned" (less than 0.9 posterior probability of belonging to any individual) or "doublets" (containing mixed genotypes) by Vireo were discarded. On average, 92% of the cells in each pool were unambiguously assigned to a single individual in the cohort (Supplementary Figure 2).

**Cell cycle scoring—**After quality control, the number of unique molecular identifiers (UMIs) mapping to each gene in each single cell were normalized for library size and log-transformed using scanpy's default normalization parameters[56]. Next, a publicly available list of cell cycle genes[58] was used in combination with scanpy to perform cell cycle scoring and assign cells to their respective stage of the cell cycle.

**Exploratory data analysis and removal of cellular contaminations—**We performed exploratory analysis at each experimental timepoint independently. Cells collected at the same timepoint were first loaded into scanpy, where normalized log-transformed UMI counts were used to identify highly variable genes. Between 701 and 1,668 highly variable genes were detected at each timepoint (mean = 1,301). Only highly variable genes were used as a basis for the remaining analyses in this section.

Technical covariates (cell culture batch) and unwanted sources of biological variation (i.e. number of UMIs per cell, proportion of reads mapping to mitochondrial genes, cell cycle scores, and reported sex) were regressed out using scanpy's regress_out() function. Next, log-UMI counts were scaled (setting 10 as the maximum value) and used as an input for principal component analysis. The first 40 principal components were used to build a k-nearest neighbours (kNN) graph (with k=15), which was used as an input for embedding and visualization with the uniform manifold approximation and projection (UMAP) algorithm[21]. This kNN graph was further used for unsupervised clustering using the Leiden algorithm[59].

At this stage, cell clustering revealed a low proportion of three contaminating cell types which were consistently detected at each timepoint: B cells, CD8+ T cells, and antigen-presenting cells (APCs). Furthermore, two additional sources of contamination (SOX4+ precursor cells, and cells expressing hallmarks of cell culture stress) were detected at zero hours of activation (Supplementary Figure 3). Cell contaminations were removed from the data set, resulting in 655,349 (91.86% of total) high quality cells kept and successfully annotated as CD4+ T cells.

**Identification of a lowly active T cell subpopulation**—Having removed cellular contaminations, highly variable genes were recalculated and the analysis described in the previous section (i.e. batch regression, scaling, principal component analysis (PCA), graph construction, embedding, and clustering) was repeated using CD4+ T cells only. Cells sampled at 16 h and 40 h showed a clear separation into two groups, one of which expressed a significantly lower number of genes and showed comparatively lower levels of previously described T cell activation markers[19] (Supplementary Figure 4a). This population of lowly active cells was separated from its original timepoint and treated as an independent group for clustering.

**Clustering and cluster annotation**—Unsupervised clustering was applied independently to the five cell groups of cells identified in the study (resting, lowly active, 16 hours, 40 hours, and five days) based on their respective kNN graphs and using the Leiden algorithm[59]. This resulted in 51 cell clusters. The similarity of these clusters to each other was assessed by performing PCA on the full data set (i.e. all cells) and estimating the Euclidean distance between pairs of clusters (from cluster center to cluster center) based on the first 100 principal components. Clusters with high levels of similarity or overlapping biological characteristics were merged together (Supplementary Figure 5b). This resulted in 38 distinct groups of cells. Gene markers for each of these groups were identified using scanpy's built-in function for gene ranking, which uses a T test to compare the average expression of a gene in a cluster versus its expression outside the cluster. Each cell group was annotated by comparing its inferred marker genes with known cell type markers reported in the literature.

**Ordering of cells in a pseudotime trajectory**—To perform trajectory inference, raw gene expression measurements for all CD4+ T cells in the study (i.e. 655,349 cells spanning all timepoints) were imported into R (v3.6.1) and analyzed using monocle3 (v0.2.0)[32]. As opposed to other analyses, where cells from each timepoint were treated independently, here some unwanted sources of variation such as cell cycle scores correlated with the biological process of interest (i.e. T cell activation). Thus, we implemented a hierarchical batch regression approach, where cell cycle scores were first regressed within each timepoint, followed by batch regression in the full data set. In brief, PCA was performed based on all cells using monocle3's PCA implementation. Next, a matrix containing the first 100 principal component coordinates for each cell was split by timepoint. Cell cycle effects were then regressed from each sub-matrix independently using limma's lmFit function[60]. Finally, these cell cycle-corrected matrices were merged back into a full PCA matrix and cell culture batch effects were regressed based on the full data set using the mutual nearest neighbors (MNN) algorithm[61].

After batch correction, the first 100 principal components were used to build a kNN graph and this graph was embedded into a two-dimensional space using UMAP. Finally, UMAP coordinates were used to infer a branched pseudotime trajectory using monocle3's learn_graph function. To identify genes that changed as a function of pseudotime, monocle3's graph test was applied to all genes. This test assesses whether cells adjacent in the trajectory show more correlated expression of a gene than cells which are far apart

(i.e. autocorrelation). Correction for multiple testing was performed using the Q value procedure[62]. A gene was considered as significantly associated with pseudotime if it had a Q value = 0.05 and a Moran's I (a measurement of the magnitude of autocorrelation) larger than 0.05[63].

**Co-expression network analysis**—Co-expression networks were created using the weighted gene co-expression network analysis (WGCNA) package (v1.69). For more details please see the Supp. Note.

**Mapping of eQTLs**—For each gene we calculated mean expression per cluster per donor. To ensure the high quality eQTL mapping, we only kept genes with non-zero expression in at least 10% of donors and mean count per million (cmp) higher than one. We retained between 8,940 and 11,516 genes. To identify cis-eQTLs we used tensorQTL (v1.0.3)[67] to run a linear regression for each SNP-gene pair, using a 500 kilobase window within the transcription start site (TSS) of each gene (i.e. cis_nominal mode). We regressed the first 15 gene expression principal components from this analysis, so as to capture the confounders within our dataset. To correct for the number of association tests performed per gene, we used a cis permutation pass per gene with 1,000 permutations. Finally, to correct for the number of genes tested and identify significant eGenes we performed a q-value correction [68] for the top associated SNP-gene pair, setting a q-value threshold of 0.1.

**Analysis of eQTL sharing across cell types**—To assess the sharing between eQTLs, we performed a meta-analysis across cell types and cell states using the multivariate adaptive shrinkage (mashR) method[25]. Please see the Supp Note.

**Modelling eQTL effect sizes as a function of network centrality**—The effect size of each gene's lead eQTL variant was modelled as a function of the gene's centrality value in the co-expression network described above. This was first done assuming a linear relationship. However, substantial heteroskedasticity was observed, which suggested a non-linear relationship, as confirmed using a Breusch-Pagan heteroskedasticity test[69]. Thus, we log-transformed the eQTL effect sizes, which resulted in homoskedastic data and a strong linear relationship between the variables. All linear models were built and tested using base R's lm() function.

**Allelic Fold change computation**—To further verify the relationship between a gene's genetic regulation and network centrality, we calculated the aFC according to Mohammadi et al.[30] using publicly available software (https://github.com/secastel/aFC)

**Modelling of dynamic psudotime-dependent eQTL effects**—To identify pseudotime-dependent eQTL effects, we divided the activation trajectory into 10 windows containing roughly equal numbers of cells (i.e. pseudotime deciles) and averaged the expression of each gene per individual within each window. To facilitate the interpretation of coefficients, pseudotime windows were scaled from 0 to 1 prior to this analysis. In order to account for the higher correlation in expression values derived from the same individual at multiple pseudotime windows, we applied linear (1) and quadratic (2) mixed models, with individuals modelled as random intercepts. We used these models to test for a significant

interaction between genotypes (i.e. the genetic dosage carried by each individual at the lead eQTL variant for that gene) and pseudotime, as follows:

1. Z_score ~ Genotype + Pseudotime + Cell_culture_batch + Sex + Age + Genotype*Pseudotime + (1 | Donor)

2. Z_score ~ Genotype + Pseudotime + Pseudotime$^2$ + Cell_culture_batch + Sex + Age + Genotype*Pseudotime + Genotype*Pseudotime$^2$ + (1 | Donor)

In both cases, the null model was computed using the same parameters while excluding the Genotype*Pseudotime and Genotype*Pseudotime$^2$ terms. P-values were calculated by comparing each model to its respective null model using analysis of variance (ANOVA). All models were implemented in R using the *lmer()* function. In order to reduce the burden imposed by multiple testing, we only applied this approach to variants previously identified as significant lead eQTL variants for a gene by tensorQTL in at least one timepoint. This was done separately for naive and memory T cells.

To ensure that the method is robust, we permuted the pseudotime windows per donor and tested for an interaction between genotype and pseudotime. A similar permutation has previously been used to test for an interaction effect between a drug and an eQTL [70]. Briefly, as the genotypes remain fixed, this strategy maintains eQTL effects while disrupting the interaction between genotype and pseudotime. By permuting the pseudotime windows 100 times (this generates a random distribution of pseudotime windows), we tested how often a dynamic eQTL would be detected in each permutation. If a test was well calibrated one would not expect to observe a large proportion of significant effects in the permuted data. Of the 7,105 and 6,304 significant static gene-SNP pairs from naive and memory T cells, respectively, we observed on average 92 and 90 significant dynamic eQTLs per each permutation round. In contrast, the number of detected dynamic eGenes in our analysis was 1,475 in naive and 1,551 in memory T cells.

**Estimation of pairwise linkage disequilibrium (LD)**—We performed LD calculations based on the individual-level genotype information for the individuals in this study obtained from genotyping. Please see the Supp Note.

**Integration of eQTLs with GWAS signals**

**Pre-processing of GWAS summary statistics:** Full summary statistics files from previous GWAS studies were downloaded from the GWAS catalogue[72]. GWAS is processed as described in Supp Note.

**Colocalization analysis:** Genomic loci of interest were identified by intersecting eQTL signals in each cell type with GWAS loci for 13 immune-mediated diseases. For each trait-cell type pair, we applied colocalization to any locus where a lead variant for a significant eQTL (q value < 0.1) was located within 100 kb and in high LD ($r^2$ > 0.5) with a significant GWAS variant (i.e. any GWAS variant with nominal p value < 1 x 10$^{-5}$, which enabled us to capture suggestive association signals). In addition, we required at least 50 variants to be available for testing at each candidate locus. At each of these loci, coloc (v4.0.4) was used to test for colocalization between the eQTL and the GWAS signals. Importantly, these

analyses were based on the recently developed masking approach, which relaxes coloc's previous assumption of a single causal variant per locus[35]. This is similar to performing conditional analyses at each locus. In brief, we defined a 500 kb window centered at the lead eQTL variant and tested for colocalization using all common variants located in the window and present in both the eQTL and the GWAS summary statistics. We used the pairwise LD calculations from our cohort as a basis for masking, setting an $r^2$ threshold of 0.01 to separate independent signals. Coloc's prior parameters were set to their recommended values in the most recent publication[35] ($p1=1 \times 10^{-4}$, $p2=1 \times 10^{-4}$, $p12=5 \times 10^{-6}$). Significant colocalizations were defined as any instances where the estimated posterior probability of a shared causal variant (PP4) was = 0.8. In order to discard potential false positives due to noisy association signals, we only kept for further analysis traits with more than one significant colocalization (11 out of 13 traits).

To infer the relationship between gene expression and disease risk at each locus, we estimated the GWAS and eQTL effect sizes (i.e. $\log_e$OR and gene expression Z-score) for the GWAS variant in highest LD with the lead eQTL variant at the locus. We concluded that a variant increased disease risk via an increase in gene expression if the variant had the same direction of effects in both studies. In the opposite case, we concluded that the variant increased disease risk via a decrease in gene expression. If the same variant had different estimates of eQTL effect size in different T cell populations, we required that all effect sizes had the same direction.

## Supplementary Material

Refer to Web version on PubMed Central for supplementary material.

## Acknowledgements

This work was funded by the Open Targets grant (OTAR040) awarded to GT. This research was funded in whole, or in part, by the Wellcome Trust [Grant number WT206194]. For the purpose of open access, the author has applied a CC BY public copyright licence to any Author Accepted Manuscript version arising from this submission. ECG is supported by a Gates Cambridge Scholarship (OPP1144). We also thank all the donors that participated in this study. We thank the Wellcome Sanger Institute Flow Cytometry facility for their assistance in cell sorting, the Sequencing facility and Cellular Genetics Informatics team for their contribution to data generation and processing. We thank Ian Dunham and Leland Taylor for critical feedback on the manuscript.

## Data availability

The raw single-cell RNA-sequencing data study is deposited in the European Genome-Phenome Archive (EGA), accession number EGAD00001008197. Genotypes are deposited in the EGA, accession number EGAD00010002291. Processed single cell data and summary statistics are available at https://trynkalab.sanger.ac.uk.

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

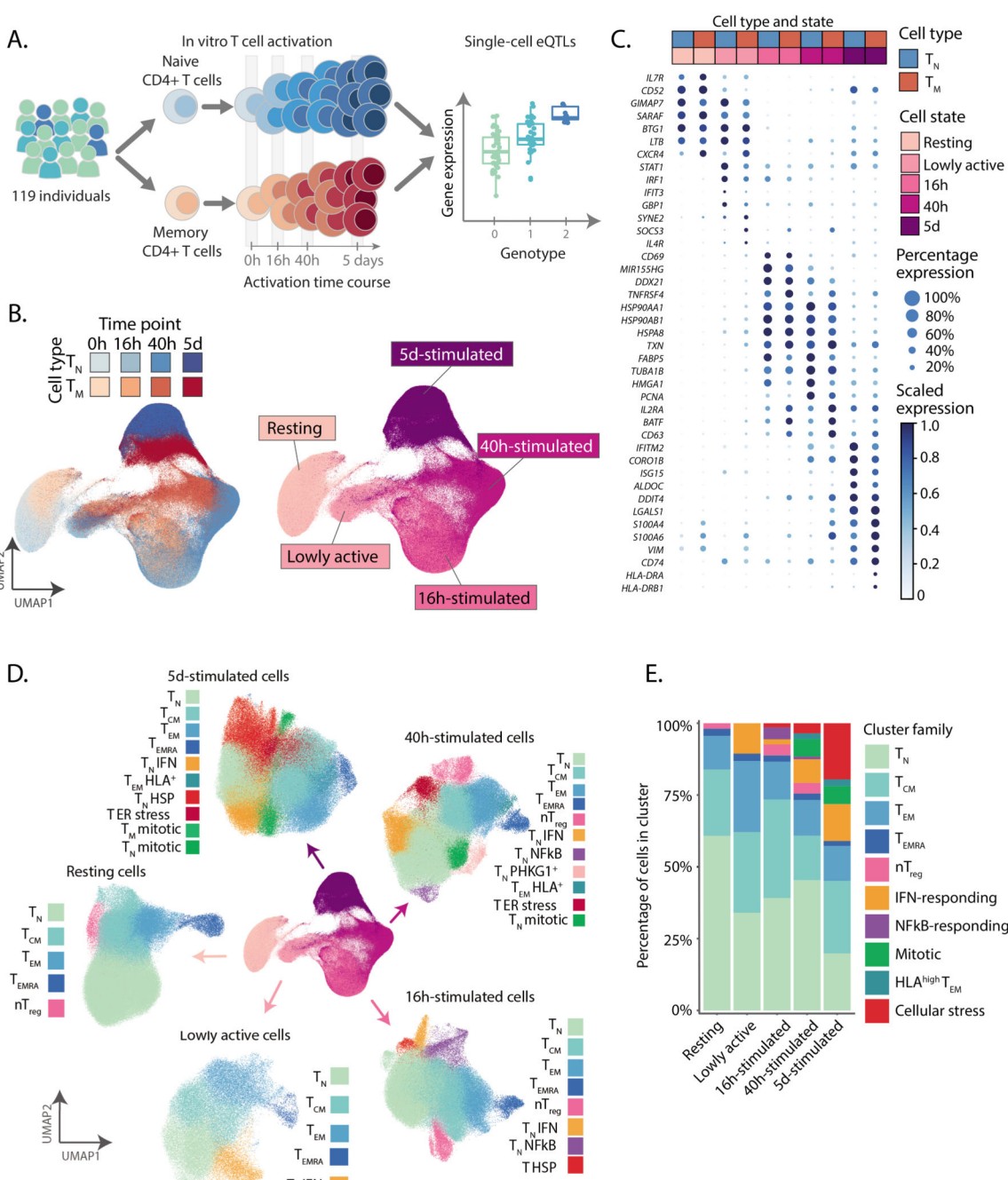

**Figure 1. A single-cell transcriptional map of CD4+ T cell activation.**
**a)** Schematic of the study design. **b)** UMAP embedding of scRNA-seq data for unstimulated CD4$^+$ T cells and at three timepoints after activation. Colors represent cell types (blue, T$_N$; red, T$_M$) and shades of colors indicate time points (lighter shades for early time points and darker shades for late time points). Right panel represents the five broad cell states. **c)** Dotplot of highly variable gene expression throughout T cell activation. Shades of blue represent average expression in each cell population, and dot sizes represent the proportion of cells expressing the gene. **d)** Separate UMAP embeddings for the five broad cell states.

Colors represent cell populations derived from unsupervised clustering. **e)** Proportion of different cluster groups present at each time point. Cell populations defined from clustering were classified into one of 10 families, represented in different colors.

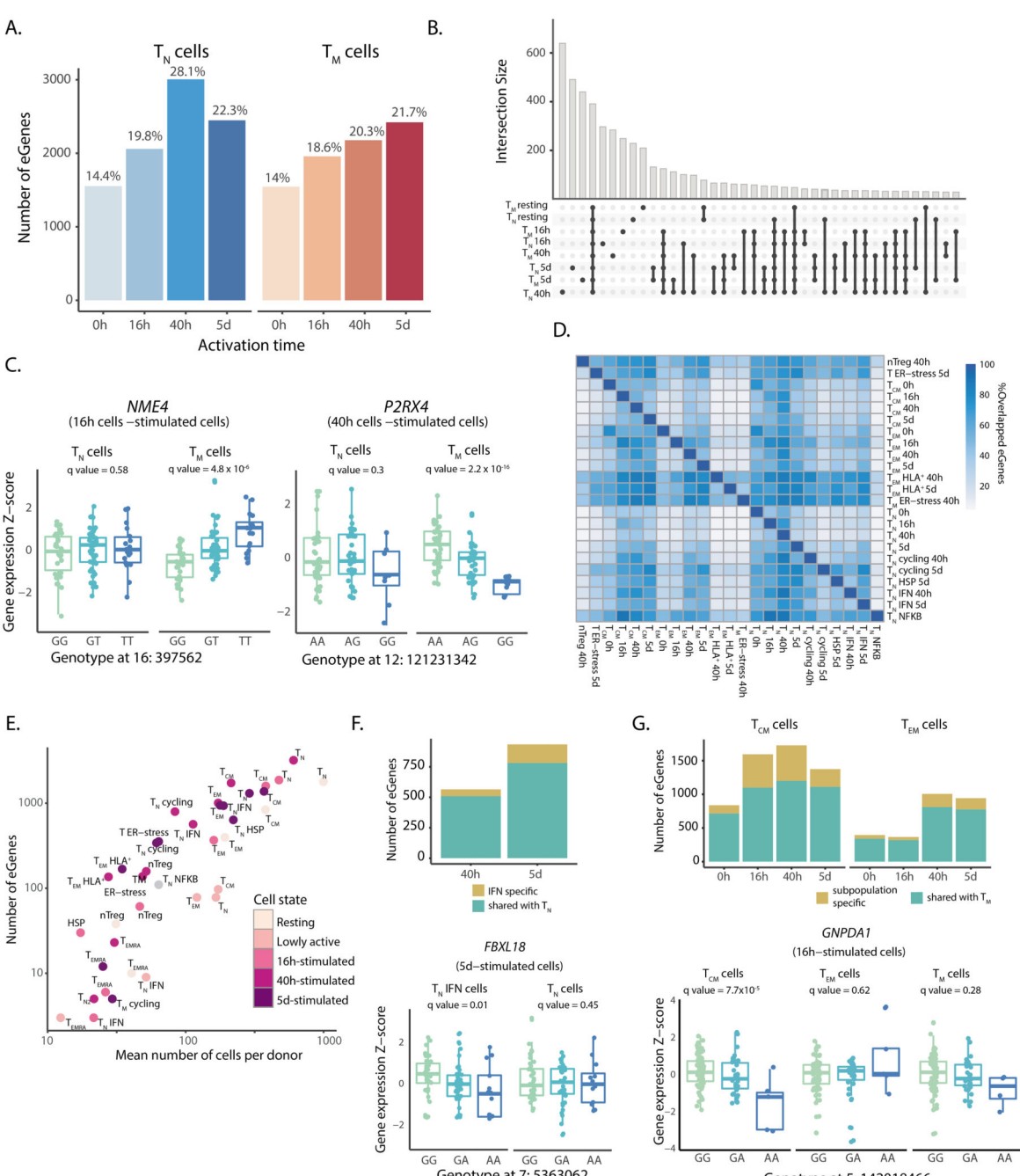

**Figure 2. eQTL mapping in resting and activated CD4+ T cells.**
**a)** Number of significant eGenes detected at each activation time point. Colors represent cell types (blue, $T_N$ – naïve; red, $T_M$ - memory). **b)** Number of significant eGenes shared between cells sampled at each time point. **c)** Example of T memory cell specific eQTLs detected at 16 h and 40 h. Box plots show mean expression value of the gene in each sample (Z-scored), stratified by genotype. Each dot represents a measurement from a separate individual. Central lines indicate the median, with boxes extending from the 25th to the 75th percentiles. Whiskers further extend by ±1.5 times the interquartile range

from the limits of each box. N of biologically independent samples: $T_N$ *NME4*: 99, $T_M$ *NME4:* 96, TN *P2RX4:* 89, TM *P2RX4:* 89. P-values were derived using tensorQTL and corrected as described in Methods. **d)** Pairwise comparison of eGenes shared between cell subpopulation. Only subpopulations with > 100 eGenes were analyzed. **e)** Scatter plot showing the correlation between number of cells per donor and number of detected eGenes in each cluster. **f)** Subpopulation specific eQTLs detected in IFN-responsive clusters. Bar plot indicates the number of eGenes detected in the IFN-responsive subpopulation that are shared with naive T cells as a whole. Boxplots show an example eQTL specific to this subpopulation. Each dot represents a measurement from a separate individual. Central lines indicate the median, with boxes extending from the 25th to the 75th percentiles. Whiskers further extend by ±1.5 times the interquartile range from the limits of each box. N of biologically independent samples: $T_N$ IFN *FBXL18:* 96, $T_N$ *FBXL18:* 87, P-values were derived using tensorQTL and corrected as described in Methods. **g)** Number of subpopulation specific eQTLs detected in $T_{CM}$ and $T_{EM}$ cells. Bar plots indicate numbers of eGenes detected in $T_{CM}$ and $T_{EM}$ subpopulations that are shared with memory T cells as a whole. Boxplots show an example eQTL specific to the TCM subpopulation. Each dot represents a measurement obtained from a separate individual. Central lines indicate the median, with boxes extending from the 25th to the 75th percentiles. Whiskers further extend by ±1.5 times the interquartile range from the limits of each box. N of biologically independent samples: $T_{CM}$ *GNPDA1:* 100, $T_{EM}$ *GNPDA1:* 103, $T_M$ *GNPDA1*: 97. P-values were derived using tensorQTL and corrected as described in Methods.

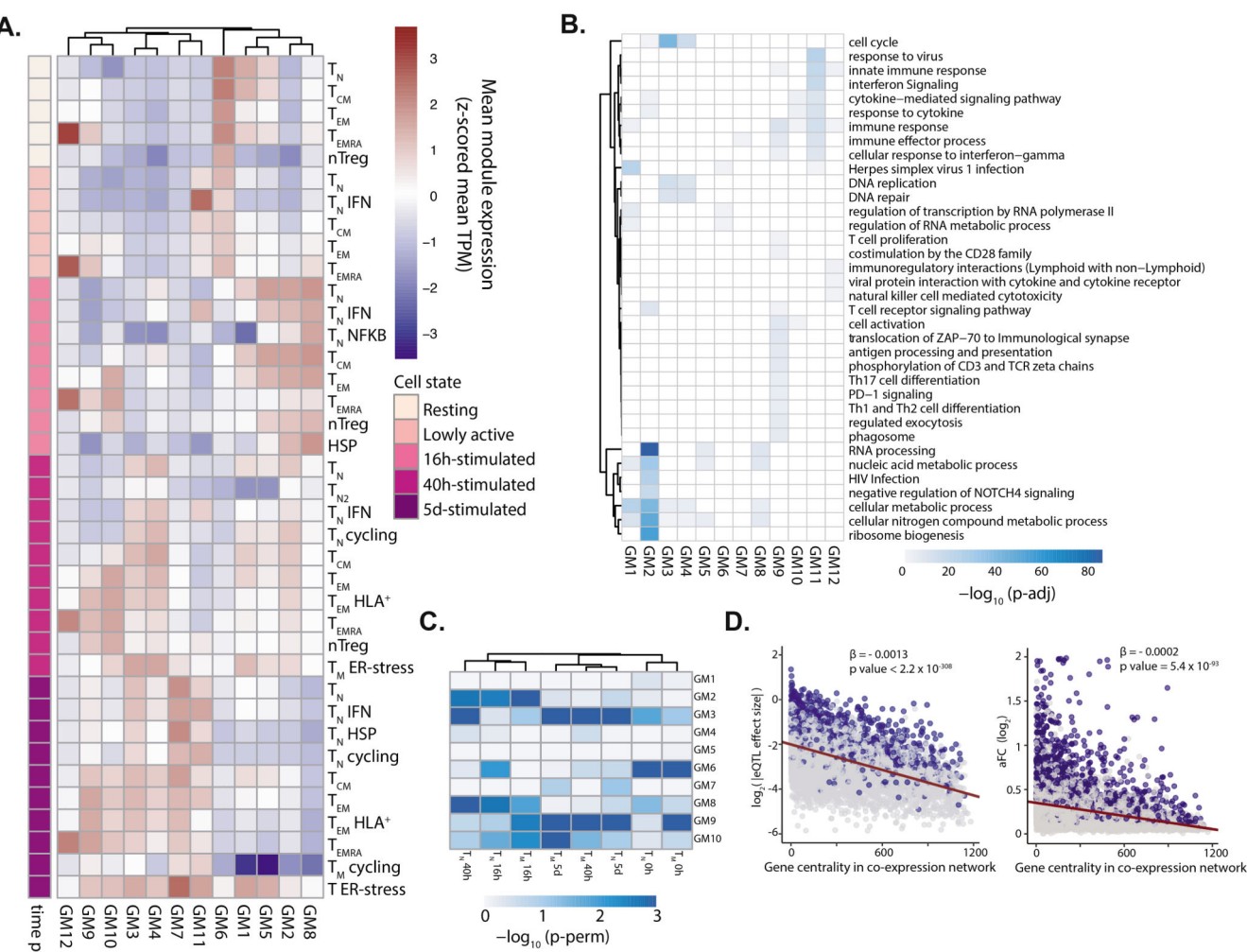

**Figure 3. eQTLs are enriched in proliferation and immune response gene modules.**
**a)** Heatmap showing the expression pattern of the 12 identified gene modules. Rows correspond to cell subpopulations. Colors represent the scaled (Z-scored) average expression of all genes belonging to a module in a given subpopulation. Gene co-expression network was built using WGCNA to identify gene modules. TPM, transcripts per million. **b)** Pathways enriched in each gene module. Shades of blue represent the log10-transformed enrichment p-values. **c)** Enrichment of eGenes in gene modules. Shades of blue represent log10-transformed p-values. P values were estimated by repeatedly permuting group labels and quantifying the proportion of times an enrichment equal to or larger than the observed one was obtained. **d)** Relationship between a gene's connectivity and the effect size of its lead eQTL variant (left) or allelic fold change (right). All eQTL effect sizes were log2-transformed. Blue dots represent significant eGenes, while gray dots represent genes which do not pass the multiple testing correction. Lines represent the best linear fits obtained from linear regression. P values were estimated by testing the null hypothesis of zero-intercepts using an F-test.

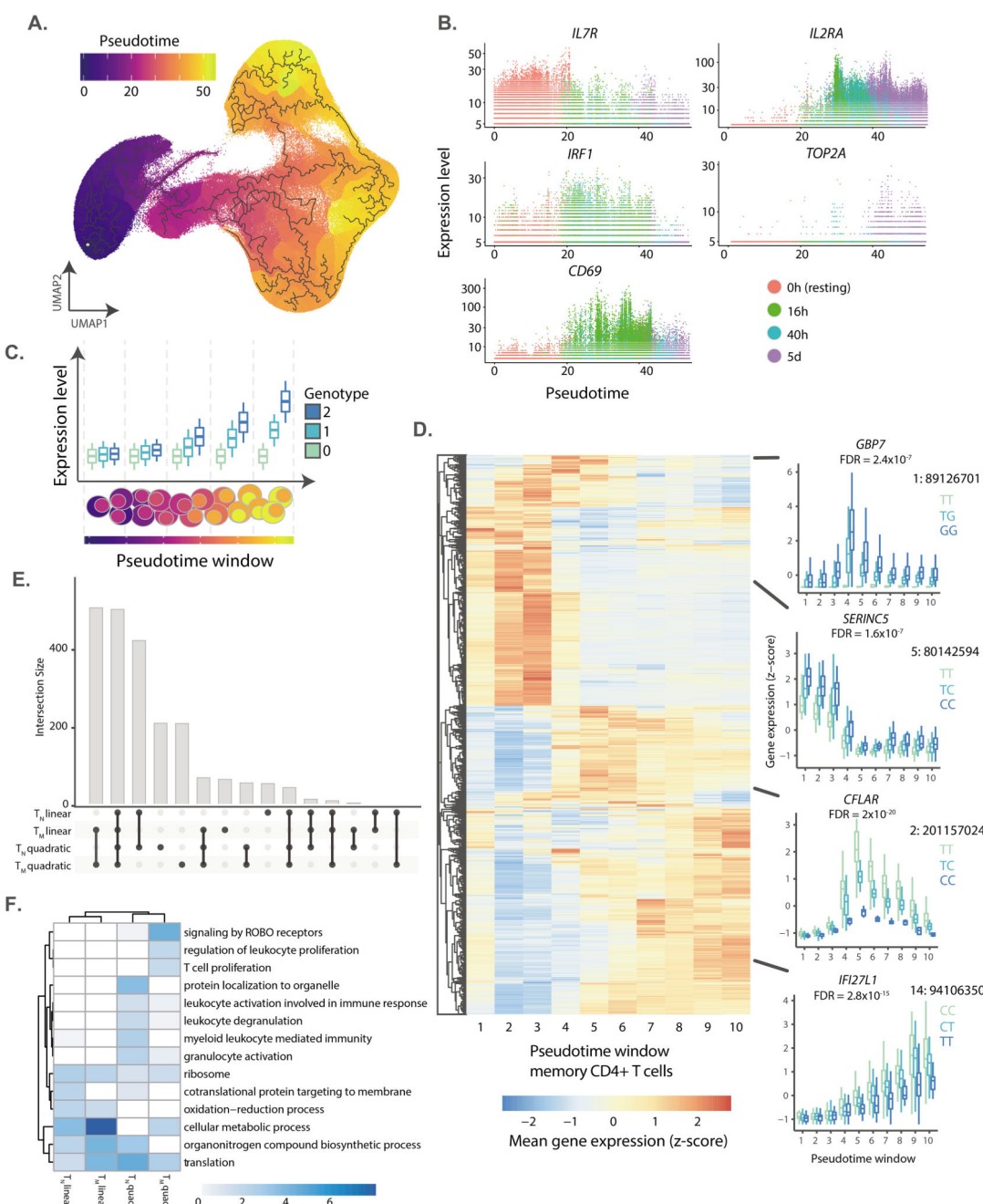

**Figure 4. eQTLs with dynamic effects during CD4+ T cell activation.**

**a)** Cells were ordered into a branched pseudotime trajectory using monocle3. The UMAP embedding shows all cells, colored by their estimated pseudotime values. Black lines indicate the inferred branched trajectory. **b)** Example genes that significantly change as a function of activation pseudotime. Each dot corresponds to a cell, and colors represent experimental time points. **c)** Schematic of the analysis approach. Cells were split into ten windows of equal cell numbers according to their estimated pseudotime values. Linear and quadratic mixed models were applied to each previously identified eGene to test for an

interaction between genotypes and T cell activation pseudotime. **d)** Heatmap showing the expression pattern on each dynamic eGene in memory T cells. Boxplots show examples of non-linear and linear dynamic eQTLs. The average expression of the gene within each pseudotime window was stratified by genotype. Central lines indicate the median, with boxes extending from the 25th to the 75th percentiles. Whiskers further extend by ±1.5 times the interquartile range from the limits of each box. N of biologically independent samples: 106. P-values were derived and corrected as described in Methods. **e)** Number of eGenes with evidence of a significant genotype-pseudotime interaction (i.e. dynamic eQTLs) in a linear or quadratic mixed model. **f)** Pathways enriched in linear and quadratic eGenes. Shades of blue represent log10-transformed enrichment p-values. Enrichment p values were estimated using a hypergeometric test and multiple testing correction was performed using the Set Counts and Sizes (SCS) method, as implemented in gprofiler2 version 0.2.0.

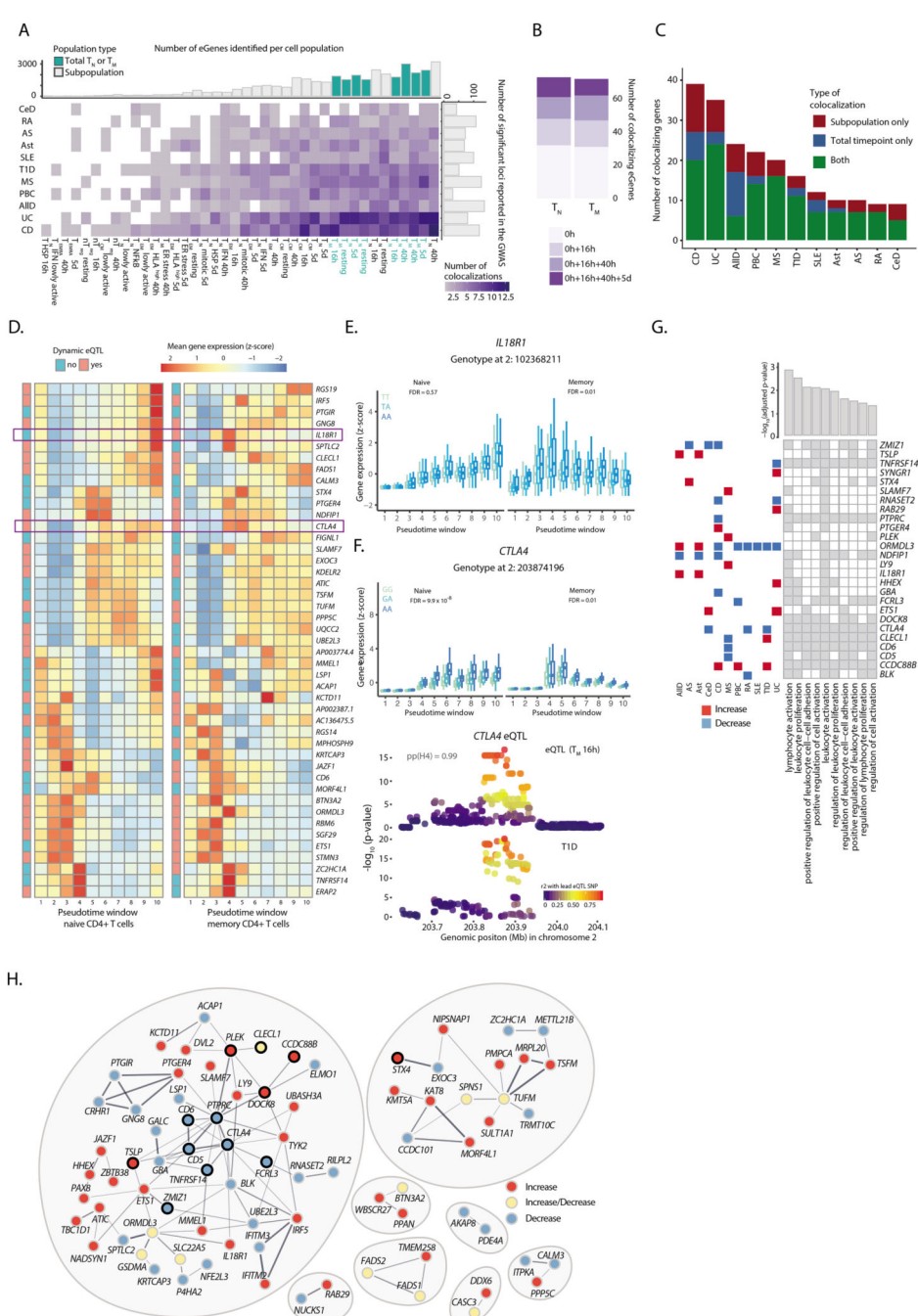

**Figure 5. Colocalization of CD4+ T cell eQTLs with GWAS associations for immune diseases.**
GWAS abbreviations: celiac disease (CeD), rheumatoid arthritis (RA), ankylosing spondylitis (AS), asthma (Ast), systemic lupus erythematosus (SLE), type 1 diabetes (T1D), multiple sclerosis (MS), primary biliary cirrhosis (PBC), allergic disease (AllD), ulcerative colitis (UC), and Crohn's disease (CD). **a)** Number of significant colocalizations between an eQTL and a GWAS signal identified for each cell type-trait combination. Marginal bar plots represent the number of independent associations reported in the GWAS (X axis) and the number of eGenes detected per subpopulation (Y axis). Light and dark bars indicate

*Nat Genet*. Author manuscript; available in PMC 2022 June 18.

whole cell populations (T_N or T_M in specific timepoint) and subpopulations, respectively. **b**) Number of additional colocalizing genes detected in stimulated cells. **c)** Number of colocalizing genes observed in whole cell populations, subpopulations or both. **d)** Heatmap showing the expression pattern of coloc eGenes in naive and memory T cells. The color of annotation boxes shows genes that are dynamic and static eQTLs. **e)** Boxplot shows *IL18R1* dynamic eQTL. The average expression of the gene within each pseudotime window was stratified by genotype. Central lines indicate the median, with boxes extending from the 25th to the 75th percentiles. Whiskers further extend by ±1.5 times the interquartile range from the limits of each box. N of biologically independent samples: 106. P-values were derived and corrected as described in Methods. **F)** Boxplot shows *CTLA4* dynamic eQTL. The average expression of the gene within each pseudotime window was stratified by genotype. Locus plot for a colocalization between a *CTLA4* dynamic eQTL and a GWAS association for type 1 diabetes. Each dot represents a variant, with colors indicating their linkage disequilibrium with the lead eQTL variant. Central lines indicate the median, with boxes extending from the 25th to the 75th percentiles. Whiskers further extend by ±1.5 times the interquartile range from the limits of each box. N of biologically independent samples: 106. P-values were derived and corrected as described in Methods. **g)** Tile plot shows enriched pathways within colocalizing genes as well as genes driving the enrichment. Barplots show adjusted p-values from the enrichment test. Squares on left show the colocalizing disease. Red, disease variant increases gene expression; blue, variant decreases gene expression. **h)** STRING network of colocalizing genes. Red, disease variant increases gene expression; blue, decreases; yellow, effect on gene expression is disease dependent. Black outline highlights genes belonging to the top enriched pathway (GO.0050867: positive regulation of cell activation).

