## [Peer Review File · Nature genetics]

Peer Review Information

Manuscript Title: Immune disease risk variants regulate gene expression dynamics during CD4+ T cell activation

Corresponding author name(s): Dr Gosia Trynka

Reviewer Comments & Decisions:

Decision Letter, initial version:
--

29th Sep 2021

Dear Gosia,

Your Article, "Immune disease risk variants regulate gene expression dynamics during CD4+ T cell activation" has now been seen by 2 referees. You will see from their comments below that while they find your work of interest, some important points are raised. We are interested in the possibility of publishing your study in Nature Genetics, but would like to consider your response to these concerns in the form of a revised manuscript before we make a final decision on publication.

As you will see, these reviews are in general quite positive. Reviewer #1 is overall supportive but has some questions about the analysis and the biological interpretation of the eQTLs and eGenes identified. Reviewer #2 is also positive and has no major concerns. The points raised by Reviewer #1 mainly require further clarifications and responses and we think that these are addressable.

We therefore invite you to revise your manuscript taking into account all reviewer and editor comments. Please highlight all changes in the manuscript text file. At this stage we will need you to upload a copy of the manuscript in MS Word .docx or similar editable format.

*1) Include a "Response to referees" document detailing, point-by-point, how you addressed each

referee comment. If no action was taken to address a point, you must provide a compelling argument. This response will be sent back to the referees along with the revised manuscript.

*2) If you have not done so already please begin to revise your manuscript so that it conforms to our Article format instructions, available [here](http://www.nature.com/ng/authors/article_types/index.html). Refer also to any guidelines provided in this letter.

[REDACTED]

We hope to receive your revised manuscript within four to eight weeks. If you cannot send it within this time, please let us know.

All the best,

Catherine

Catherine Potenski, PhD
Chief Editor
Nature Genetics
1 NY Plaza, 47th Fl.
New York, NY 10004
catherine.potenski@us.nature.com
<https://orcid.org/0000-0002-4843-7071>

Referee expertise:

Referee #1: immunology, genetics, single-cell analysis

Referee #2: genetics, immune diseases, eQTL analysis

Reviewers' Comments:

Reviewer #1:

Remarks to the Author:

This manuscript provides a highly important resource of eQTL of stimulated CD4+ T cells with single-cell, high-quality dataset. The experiment is well designed in that gene expression is quantified at multiple time points, considering complex response of CD4+ T cells to stimulation. The manuscript is statistically well analyzed. The identified colocalization of immune-mediated disease GWAS signals with dynamic eQTLs is a very insightful finding in this field. Yet, the number of eQTLs identified has been relatively small, and I think there are several points which need to be addressed:

1. Page 7: "We observed that between 210 and 640 eGenes were only detected in individual cell states (Figure 2B). For example, kinase NME4 and purinoceptor P2RX4 were only detected as eGenes in memory T cells at 16h and 40h of activation, respectively (Figure2C)." "Finally, to understand which cell functions might differ in their genetic regulation across cell types, we tested for pathway enrichment in eGenes detected in one cell type and not the other, i.e. in memory but not naïve T cells and vice versa (Figure 2D)."

The power for identifying eQTLs greatly depends on gene expression level. My consideration is whether these cell state-specific eGenes are the consequence of cell state-dependent gene regulation or cell state-specific gene expression. For me, pathway analysis of eGenes doesn't make much sense if most of the genes have eQTLs as GTEx reported (The GTEx consortium, Science. 2020). Detectability of eGenes are solely relying on statistical power due to sample size, gene expression level, allele frequency, etc. I'd like to see the difference in the result of pathway analysis between cell state-specific eGenes and cell state-specific expressed genes, and to be explained about the biological meaning of pathway analysis with cell state-specific eGenes.

In addition, to see the merit of single cell-based eQTL analysis, I'd like to see the examples of eQTLs that were detected only in this dataset at specific cell state, compared to previously reported immune cell eQTLs.

2. Page 12: "Finally, we observed that eQTL effect sizes negatively correlated with the centrality

values of the corresponding eGenes in the coexpression network, i.e. eGenes with larger eQTL effects were more likely to be less connected in the network (Figure 3D). This suggests that genes at the edges of the co-expression network are more tolerant to variability in gene expression.”

Although this finding is very interesting, biological interpretation of eQTL beta is not straightforward as previously proposed (Mohammadi, et al, Genome Res. 2017). Allelic fold change, introduced by this group, might be preferable to using log-transformed effect size in this comparison.

3. Page 13: “We identified 2,265 genes with dynamic eQTL effects, which comprised 34% of eGenes in our dataset (Supplementary Table 5). We applied both linear and quadratic models and observed that most eQTLs followed linear dynamics across the activation trajectory (74% and 76% in naive and memory T cells, respectively) (Figure 4E).”

An important feature of this paper is that it examines the dynamically changing gene expression of T cells after stimulation at the single cell level. This data set has the potential to not only identify eQTLs, but also to get closer to the regulatory mechanism of eQTLs. Several researchers analyzed the interaction of eQTL effect size and genome-wide gene expression that are dependent on intrinsic or extrinsic factors (context-dependent eQTL) (Zhernakova et al. Nat Genet. 2017;49:139-145.; Ota, et al Cell. 2021;184:3006-3021.e17.). Similar analysis was also successful in scRNAseq data of PBMC from 45 donors (van der Wijst et al. Nat Genet. 2018;50:493-497.). Therefore, context-dependent eQTL analysis in this dataset may reveal relationships between several eQTLs/eGenes and functional gene modules to deepen the understanding of eQTL regulation in stimulated T cells.

4. Page 13: “We identified 2,265 genes with dynamic eQTL effects, which comprised 34% of eGenes in our dataset (Supplementary Table 5).”

I'd like to see the analytic stability of dynamic eQTL analysis. How was the distribution of p values? If possible, permutation test would be one option to ensure the robustness of the results.

Reviewer #2:

Remarks to the Author:

Soskic and colleagues investigate how gene expression and genetic influences thereon can vary when cells are placed in different contexts, and how some of these context-specific effects are relevant to disease risk. They use the well-established model of T lymphocyte activation by CD3/CD28, profiling naïve and memory CD4+ T cells from 119 healthy donors at baseline and at 16 hours, 40 hours and 5 days after activation. They generate single-cell RNAseq data on 655,349 individual cells across these individuals and conditions. They capture both stable and transient sub-populations of T cells, recapitulating what we know about the process of activation. They identify eQTLs across these cell populations, and show that 2265/6407 are only detected in the transient populations, indicating that gene regulation is dynamic over activation. They then show that these context-specific eQTLs overlap immune-mediated disease associations; in fact, they account for 60% of all overlaps. This suggests many disease risk variants alter gene regulation during dynamic processes rather than altering baseline expression.

Overall, this is a well-written paper with an emphasis on the importance of context-specific studies of immune cells. It is a very instructive application of scRNAseq to a more focused problem, showing that

cell dynamics can be dissected with great precision (rather than coarser studies looking at all lymphocytes, for instance). The work is timely, as a major issue in human genetics is trying to understand how disease risk variants influence biology – and why so few associations seem to overlap eQTLs, despite the huge enrichment of heritability on gene regulatory elements. Soskic et al provide at least a partial answer to this: they are context-specific eQTLs not observed in resting cells. I have no major questions or concerns about the work presented, but only a few minor comments:

- Figures 2b/4e: some more explanation in the legend on how to decode the dot diagram would be useful
- I am a little puzzled by the differences in colocalization between single-cell and pseudo-bulk analyses (Figure 5c). Are these likely to be false positives (coloc has a somewhat elevated false positive rate), or is this a detection issue?
- The authors document how their analyses recapitulate known T cell activation biology through the paper. To my eyes, it makes it a little challenging for the reader to follow all the diverse aspects. The authors may want to pare back some of these examples.

Author Rebuttal to Initial comments

Reviewers' Comments:

We thank all the Reviewers for their feedback. We have addressed all of the comments and we believe this has greatly improved our manuscript.

Reviewer #1:

Remarks to the Author:

This manuscript provides a highly important resource of eQTL of stimulated CD4+ T cells with a single-cell, high-quality dataset. The experiment is well designed in that gene expression is quantified at multiple time points, considering the complex response of CD4+ T cells to stimulation. The manuscript is statistically well analyzed. The identified colocalization of immune-mediated disease GWAS signals with dynamic eQTLs is a very insightful finding in this field. Yet, the number of eQTLs identified has been relatively small, and I think there are several points which need to be addressed:

We thank the Reviewer for these positive comments and for highlighting the importance of our work.

1. Page 7: *“We observed that between 210 and 640 eGenes were only detected in individual cell states (Figure 2B). For example, kinase NME4 and purinoceptor P2RX4 were only detected as eGenes in memory T cells at 16h and 40h of activation, respectively (Figure 2C) [...] Finally, to understand which cell functions might differ in their genetic regulation across cell types, we tested for pathway enrichment in eGenes detected in one cell type and not the other, i.e. in memory but not naïve T cells and vice versa (Figure 2D).”*

The power for identifying eQTLs greatly depends on gene expression level. My consideration is whether these cell state-specific eGenes are the consequence of cell state-dependent gene regulation or cell state-specific gene expression. For me, pathway analysis of eGenes doesn't make much sense if most of the genes

have eQTLs as GTEx reported (The GTEx consortium, Science. 2020). Detectability of eGenes is solely relying on statistical power due to sample size, gene expression level, allele frequency, etc. I'd like to see the difference in the result of pathway analysis between cell state-specific eGenes and cell state-specific expressed genes, and to be explained about the biological meaning of pathway analysis with cell state-specific eGenes.

In addition, to see the merit of single cell-based eQTL analysis, I'd like to see the examples of eQTLs that were detected only in this dataset at specific cell states, compared to previously reported immune cell eQTLs.

We thank the Reviewer for their valuable feedback. Upon a reflection we agree with this comment, indeed, in a study with sufficient statistical power every gene would be expected to have at least one eQTL, and therefore pathway enrichment would be meaningless. We thus removed this analysis from our paper. We believe this does not impact the conclusions of our study.

Regarding detection of cell state specific effects, we have first performed a global comparison of our eQTL signals with two large bulk RNA-seq studies that also included CD4+ T cells: BLUEPRINT and DICE (Chen et al. 2016; Schmiedel et al. 2018). We compared our eQTLs to gene-variant pairs available in the published summary statistics of these studies for any CD4+ T cell populations. Comparisons were made in terms of the direction and magnitude of eQTL effects using the MashR method (Urbut et al. 2018). We identified the lead variant for each gene and we randomly selected four variants per gene as the random set to fit the mashr mode.

We have now added these results to our manuscript, summarising them as follows: “*We observed a high degree of replicability, with previously published eQTLs (0.67-075); **Supplementary Figure 8C***). However, *eQTL sharing was reduced when taking into account both the direction and the magnitude of eQTL effects (0.28-0.34); **Supplementary Figure 8C***), suggesting that effect sizes might differ between different transcriptomics profiling strategies, naive and memory cells, and across T cell activation timepoints.”

C.

Supplementary Figure 8. eQTL sharing. C) Overlaps of eQTLs with DICE and BLUEPRINT were assessed by comparing the effect sizes and directions of eQTLs across cell types (naive and memory cells) and time points using mashR.

In addition, the Nathan et al., study (Nathan et al 2021 bioRxiv) also demonstrated that if the eQTL is significant in both their scRNAseq and the public bulk data, it largely has the same direction of an effect. We unfortunately do not have access to their raw data to recalculate the replication rate and compare it to our calculations.

Furthermore, by comparing our activation eQTLs to previously reported results in any of DICE's CD4+ T cell populations, we identified many examples of eQTLs present in activated T cells, but not detected in resting cells. For example, *CD6* has a strongly significant effect in our data, an FDR-adjusted p-value of

5.7×10^{-6} in stimulated memory T cells, while it is not reported as an eQTL in the DICE dataset. *CFLAR* is the strongest dynamic eQTLs in our study (**Figure 4D**), but is not identified as an eQTL in DICE.

Next we specifically limited ourselves to demonstrating the value of single cell eQTL mapping by comparison to the pseudo-bulk eQTLs within our dataset. To address the Reviewers comment and provide examples of genes detected at single cell level, we asked if eGenes that were only observed at the single cell level in our study, i.e. significant in cell subpopulations but not in the whole cell type, were also significant in published bulk RNAseq data. In the main text we now provide examples of such genes that demonstrate eQTL effects in our dataset at a single cell level but are not detected in the DICE data:

Page 5: For example, *VAMP8* and *AIMP1* eQTLs which are T_{CM} specific in our dataset ($p_{adj} = 5 \times 10^{-4}$ and $p_{adj} = 1.04 \times 10^{-4}$, respectively) and *RNF168* specific to IFN expressing cell cluster ($p_{adj} = 6.2 \times 10^{-3}$), were not detected in any of the T cell populations in DICE. These data suggest that further increase in power to detect cluster eQTLs will likely result in many more eQTLs that were not previously identified in bulk cell types and tissues.

2. Page 12: “Finally, we observed that eQTL effect sizes negatively correlated with the centrality values of the corresponding eGenes in the coexpression network, i.e. eGenes with larger eQTL effects were more likely to be less connected in the network (Figure 3D). This suggests that genes at the edges of the co-expression network are more tolerant to variability in gene expression.”

Although this finding is very interesting, the biological interpretation of eQTL betas is not straightforward as previously proposed (Mohammadi, et al, Genome Res. 2017). Allelic fold change, introduced by this group, might be preferable to using log-transformed effect size in this comparison.

As suggested by the Reviewer, we added an analysis using allelic fold changes (aFC). We observed a similar association between aFCs and network centrality values, as shown in **Figure 3D** and **Figure S9**. We have included these observations in the **Results** section (Page 6) of our revised manuscript.

Page 6: “Finally, we observed that eQTL effect sizes, as well as log-transformed allelic fold-changes³⁰ negatively correlated with the centrality values of their corresponding eGenes in the coexpression network, i.e. eGenes with larger eQTL effects were less connected in the network (**Figure 3D** and **Supplementary Figure 9B-C**). This suggests that genes at the edges of the co-expression network are more tolerant to variation in gene expression.”

D.
Figure 3. D) Relationship between a gene's connectivity and the effect size of its lead eQTL variant (left) or allelic fold change (right). All eQTL effect sizes were log₂-transformed. Blue dots represent significant eGenes, while gray dots represent genes which do not pass the multiple testing correction.

Supplementary Figure 9. Identification of gene expression patterns active throughout CD4⁺ T cell activation. **B)** Relationship between a gene's connectivity (as inferred from co-expression network analysis) and the effect size of its lead eQTL signal. All eQTL effect sizes were log-transformed. Blue dots represent significant eGenes, while gray dots represent genes which do not pass the eQTL multiple test correction. **C)** Relationship between a gene's connectivity and the allelic fold change (aFC) of its lead eQTL signal. All eQTL effect sizes were log-transformed.

Blue dots represent significant eGenes, while gray dots represent genes which do not pass the eQTL multiple test correction.

3. Page 13: “We identified 2,265 genes with dynamic eQTL effects, which comprised 34% of eGenes in our dataset (Supplementary Table 5). We applied both linear and quadratic models and observed that most eQTLs followed linear dynamics across the activation trajectory (74% and 76% in naive and memory T cells, respectively) (Figure 4E).”

An important feature of this paper is that it examines the dynamically changing gene expression of T cells after stimulation at the single cell level. This data set has the potential to not only identify eQTLs, but also get closer to the regulatory mechanism of eQTLs. Several researchers analyzed the interaction of eQTL effect size and genome-wide gene expression that are dependent on intrinsic or extrinsic factors (context-dependent eQTL) (Zhernakova et al. Nat Genet. 2017;49:139-145.; Ota, et al Cell. 2021;184:3006-3021.e17.). Similar analysis was also successful in scRNAseq data of PBMC from 45 donors (van der Wijst et al. Nat Genet. 2018;50:493-497.). Therefore, context-dependent eQTL analysis in this dataset may reveal relationships between several eQTLs/eGenes and functional gene modules to deepen the understanding of eQTL regulation in stimulated T cells.

We thank the Reviewer for suggesting this analysis. Following the framework set out in van der Wijst et al. Nat Genet 2018, we performed co-expression QTL analysis to test if a SNP associated with expression of an eGene also affects its co-expression with other genes. To increase the likelihood of finding significant associations, we analysed each time-point/cell-type combination separately. Furthermore, we restricted the analysis to SNP-eGene pairs which: 1) colocalize with immune disease risk loci, and 2) are highly variable between clusters, assessing the genes' co-expression relationship with all other highly variable genes identified in that condition. We used Spearman's rank correlation to calculate the co-expression between all pairs of genes and fitted a weighted linear model to assess the strength of this correlation with the genotype for the SNP in question, supplying the square root of the number of cells per donor as weights. The median number of cells per donor was 746 (range: 53-2142). We identified a total of 10,644 co-expression QTLs (co-QTLs) for 45 genes, with a median of 56 co-QTLs per gene (range: 1-951). Of all the eGenes tested, 3-7 genes per condition had at least one significant QTL and 2-4 genes per condition had more than 100 co-expression QTLs. The SNP associated with *ERAP2* had the highest number of significant co-expression QTLs in all six of the conditions tested (range: 815-951). In order to determine the most significant co-expression QTL per SNP-eGene pair, a permutation-based (n=100) FDR-approach was used to determine a gene-wise statistical significance threshold, whereby the most significant co-expression QTL p-value was compared to the p-values obtained when randomly shuffling the genotypes. An FDR < 0.05 was considered as statistically significant. After permutations, 41 out of the 45 eGenes remained significant for co-expression QTLs (Figure R1).

Figure R1. Number of significant co-expression QTLs detected per cell type and time point.

However, we were not confident that these eQTLs represented real signals. For example *PTGIR* and *SOCS1* were identified as significant ($p\text{-value} = 5 \times 10^{-5}$, $FDR = 0.026$), but visualisation of the relationship between their expression levels stratified by genotype does not show an evident separation and is strongly affected by random fluctuations and drop outs (**Figure R2**). We have manually expected all significant genes and observed similar confounders with the majority of identified relationships.

Figure R2. Co-expression patterns of *PTGIR* and *SOCS1* stratified by genotype at a nearby co-expression QTL

Thus, while we agree with the reviewer that this is a very interesting question to address, we think that at present this analysis is not feasible due to the high amount of noise and drop outs inherent to single-cell RNA-seq. To overcome these confounders we would need a bigger cohort and deeper sequencing to more confidently detect significant coexpression QTLs.

4. Page 13: “*We identified 2,265 genes with dynamic eQTL effects, which comprised 34% of eGenes in our dataset (Supplementary Table 5).*”

I'd like to see the analytic stability of dynamic eQTL analysis. How was the distribution of p values? If possible, permutation test would be one option to ensure the robustness of the results.

We thank the Reviewer for raising this important point, which we have addressed by developing a permutation approach, where we permuted the pseudotime windows per donor and tested for an interaction between genotype and pseudotime. A similar permutation has previously been used to test for an interaction effect between a drug and an eQTL (Davenport et al. 2018). Briefly, as the genotypes remain fixed, this strategy maintains eQTL effects while disrupting the interaction between genotype and pseudotime. By permuting the pseudotime windows 100 times (this generates a random distribution of pseudotime windows), we tested how often a dynamic eQTL would be detected in each permutation. If a test was well calibrated one would not expect to observe a large proportion of significant effects in the permuted data. Of the 7105 and 6304 significant static gene-SNP pairs from naive and memory T cells, respectively, we observed on average 92 and 90 significant dynamic eQTLs per each permutation round (see Supplementary Figure 10 B for the full distribution). In contrast, the number of detected dynamic eGenes in our analysis was 1,475 in naive and 1,551 in memory T cells, an order of magnitude larger. This suggests that the method for dynamic eQTL mapping is robust, and that a genetic effect is observed only when the pseudotime order is preserved.

In addition, as suggested by the reviewer, we visualised the p-values resulting from dynamic eQTL analysis using quantile-quantile (QQ) plots. This visualisation confirmed inflation when comparing to the expectation under a uniform distribution.

We have incorporated these results into our revised manuscripts in **Figure S10**, and in the **Methods** (Page 20). We also added the following statement to the **Results** section:

To model dynamic eQTLs, we divided the pseudotime trajectory into ten bins and averaged the expression of genes per individual in each bin (**Methods**). Splitting the trajectory enabled us to control for the numbers

of cells and therefore to reliably estimate mean gene expression. We then used mixed models to test for a significant interaction between genotype and average pseudotime value per bin (**Figure 4C** and **Methods**). This enabled us to identify eQTLs for which the effect size changed as a function of activation time. We identified 2,265 genes with dynamic eQTL effects, which comprised 34% of eGenes in our dataset (**Supplementary Table 5** and **Supplementary Figure 10A**). We used a permutation based strategy to validate that this test was well calibrated (**Methods** and **Supplementary Figure 10B**).

A.

B.

Supplementary Figure 10. Dynamic eQTL discovery A) Q-Q plots of all SNP-gene pairs that are tested in the dynamic model. B) Distribution of the number of significant dynamic eQTLs in the permuted data.

Reviewer #2:

Remarks to the Author:

Soskic and colleagues investigate how gene expression and genetic influences thereon can vary when cells are placed in different contexts, and how some of these context-specific effects are relevant to disease risk. They use the well-established model of T lymphocyte activation by CD3/CD28, profiling naïve and memory CD4⁺ T cells from 119 healthy donors at baseline and at 16 hours, 40 hours and 5 days after activation. They generate single-cell RNAseq data on 655,349 individual cells across these individuals and conditions. They capture both stable and transient sub-populations of T cells, recapitulating what we know about the process of activation. They identify eQTLs across these cell populations, and show that 2265/6407 are only detected in the transient populations, indicating that gene regulation is dynamic over activation. They then show that these context-specific eQTLs overlap immune-mediated disease associations; in fact, they account for 60% of all overlaps. This suggests many disease risk variants alter gene regulation during dynamic processes rather than altering baseline expression.

Overall, this is a well-written paper with an emphasis on the importance of context-specific studies of immune cells. It is a very instructive application of scRNAseq to a more focused problem, showing that cell dynamics can be dissected with great precision (rather than coarser studies looking at all lymphocytes, for instance). The work is timely, as a major issue in human genetics is trying to understand how disease risk variants influence biology – and why so few associations seem to overlap eQTLs, despite the huge enrichment of heritability on gene regulatory elements. Soskic et al provide at least a partial answer to this:

they are context-specific eQTLs not observed in resting cells. I have no major questions or concerns about the work presented, but only a few minor comments:

We thank the Reviewer for their encouraging feedback.

- Figures 2b/4e: some more explanation in the legend on how to decode the dot diagram would be useful

We thank the reviewer for raising this point. We have added the following description: “Each circle represents one cell state. Circles are either gray (not part of the intersection), or black (participating in the intersection).”

- I am a little puzzled by the differences in colocalization between single-cell and pseudo-bulk analyses (Figure 5c). Are these likely to be false positives (coloc has a somewhat elevated false positive rate), or is this a detection issue?

Thank you for pointing this out. To clarify this further we changed our annotation to: “*subpopulation only*” and “*Total timepoint only*”. We believe that these differences are due to a detection issue, where subpopulations in which we detect less eQTLs (**Figure 2E**) also result in less detected colocalizations.

- The authors document how their analyses recapitulate known T cell activation biology through the paper. To my eyes, it makes it a little challenging for the reader to follow all the diverse aspects. The authors may want to pare back some of these examples.

Thank you for this comment. We have removed several examples to make the text easier to follow.

Decision Letter, first revision:

Our ref: NG-A58160R

14th Jan 2022

Dear Gosia,

Thank you for submitting your revised manuscript "Immune disease risk variants regulate gene expression dynamics during CD4+ T cell activation" (NG-A58160R). It has now been seen by the original referees and their comments are below. The reviewers find that the paper has improved in revision, and therefore we'll be happy in principle to publish it in Nature Genetics, pending minor revisions to comply with our editorial and formatting guidelines.

We are now performing detailed checks on your paper and will send you a checklist detailing our

editorial and formatting requirements soon. Please do not upload the final materials and make any revisions until you receive this additional information from us.

Congratulations to you and your team on the paper!

All the best,

Catherine

Catherine Potenski, PhD
Chief Editor
Nature Genetics
1 NY Plaza, 47th Fl.
New York, NY 10004
catherine.potenski@us.nature.com
<https://orcid.org/0000-0002-4843-7071>

Reviewer #1 (Remarks to the Author):

Soskic et al. responded appropriately to the reviewer's comments. They compared their eQTLs to gene-variant pairs available in the summary statistics of BLURPRINT and DICE for any CD4+ T cell populations to find that effect sizes might differ between different transcriptomics profiling strategies, naive and memory cells, and across T cell activation timepoints. Moreover, they identified many examples of eQTLs present in activated T cells, but not detected in resting cells. They provided examples of genes that demonstrate eQTL effects in their dataset at a single cell level but are not detected in the DICE data. Although these results are limited to a subset of genes, they demonstrate the significance of single-cell eQTL analysis. Although it is unfortunate that they failed to get robust result from co-expression eQTL analysis, it is understandable that the results were not reliable due to the limitation of data accuracy of single-cell RNAseq. This manuscript reports one of the most extensive single-cell eQTL analysis at this time, and it is worth reporting.

Reviewer #2 (Remarks to the Author):

The authors have responded to all my comments

Final Decision Letter:

In reply please quote: NG-A58160R1 Trynka

30th Mar 2022

Dear Gosia,

I am delighted to say that your manuscript "Immune disease risk variants regulate gene expression dynamics during CD4+ T cell activation" has been accepted for publication in an upcoming issue of Nature Genetics.

Your paper will be published online after we receive your corrections and will appear in print in the next available issue. You can find out your date of online publication by contacting the Nature Press Office (press@nature.com) after sending your e-proof corrections. Now is the time to inform your Public Relations or Press Office about your paper, as they might be interested in promoting its publication. This will allow them time to prepare an accurate and satisfactory press release. Include your manuscript tracking number (NG-A58160R1) and the name of the journal, which they will need when they contact our Press Office.

Please note that *Nature Genetics* is a Transformative Journal (TJ). Authors may publish their research with us through the traditional subscription access route or make their paper immediately open access through payment of an article-processing charge (APC). Authors will not be required to

make a final decision about access to their article until it has been accepted. [Find out more about Transformative Journals](https://www.springernature.com/gp/open-research/transformative-journals)

Authors may need to take specific actions to achieve [compliance with funder and institutional open access mandates](https://www.springernature.com/gp/open-research/funding/policy-compliance-faqs). If your research is supported by a funder that requires immediate open access (e.g. according to [Plan S principles](https://www.springernature.com/gp/open-research/plan-s-compliance)) then you should select the gold OA route, and we will direct you to the compliant route where possible. For authors selecting the subscription publication route, the journal's standard licensing terms will need to be accepted, including [self-archiving and license to publish](https://www.nature.com/nature-portfolio/editorial-policies/self-archiving-and-license-to-publish). Those licensing terms will supersede any other terms that the author or any third party may assert apply to any version of the manuscript.

Please note that Nature Research offers an immediate open access option only for papers that were first submitted after 1 January, 2021.

If you have not already done so, we invite you to upload the step-by-step protocols used in this manuscript to the Protocols Exchange, part of our on-line web resource, natureprotocols.com. If you complete the upload by the time you receive your manuscript proofs, we can insert links in your article that lead directly to the protocol details. Your protocol will be made freely available upon publication of your paper. By participating in natureprotocols.com, you are enabling researchers to more readily reproduce or adapt the methodology you use. [Natureprotocols.com](http://natureprotocols.com) is fully searchable, providing your

protocols and paper with increased utility and visibility. Please submit your protocol to <https://protocolexchange.researchsquare.com/>. After entering your nature.com username and password you will need to enter your manuscript number (NG-A58160R1). Further information can be found at <https://www.nature.com/nature-portfolio/editorial-policies/reporting-standards#protocols>

Congratulations to you and your team on this paper!

All the best,

Catherine

Catherine Potenski, PhD
Chief Editor
Nature Genetics
1 NY Plaza, 47th Fl.
New York, NY 10004
catherine.potenski@us.nature.com
<https://orcid.org/0000-0002-4843-7071>